# Molecular Characterization of a Recombinant Isolate of Tomato Leaf Curl New Delhi Virus Associated with Severe Outbreaks in Zucchini Squash in Southern Italy

**DOI:** 10.3390/plants12132399

**Published:** 2023-06-21

**Authors:** Mariarosaria Mastrochirico, Roberta Spanò, Rita Milvia De Miccolis Angelini, Tiziana Mascia

**Affiliations:** Department of Soil, Plant and Food Sciences, University of Bari “Aldo Moro”, 70126 Bari, Italy; mariarosaria.mastrochirico@uniba.it (M.M.); roberta.spano@uniba.it (R.S.); ritamilvia.demiccolisangelini@uniba.it (R.M.D.M.A.)

**Keywords:** complete genome sequence, recombination, tomato leaf curl New Delhi virus

## Abstract

The molecular characterization of a tomato leaf curl New Delhi virus (ToLCNDV) isolate, denoted ToLCNDV-Le, is reported. The virus was associated with severe and recurrent outbreaks in protected crops of zucchini squash grown in the Province of Lecce (Apulia, southern Italy). The fully sequenced genome of ToLCNDV-Le consists of two genomic components named DNA-A and DNA-B of 2738 and 2683 nt in size, respectively. Like other ToLCNDV isolates, ToLCNDV-Le DNA-A contains the AV2 and AV1 open reading frames (ORFs) in the virion-sense orientation and five additional ORFs named AC1, AC2, AC3, AC4 and AC5 in the complementary-sense orientation. The DNA-B contains BV1 ORF in the virion-sense orientation and BC1 ORF in the complementary-sense orientation. No DNA betasatellites were found associated with ToLCNDV-Le in naturally infected samples. Phylogenetic analysis clustered ToLCNDV-Le with the ToLCNDV-ES strain of western Mediterranean Basin isolates. Consequently, the ToLCNDV-ES-[IT-Zu-Le18] name is proposed as the descriptor for ToLCNDV-Le. Using recombination detection program RDP4, one putative recombination breakpoint (Rbp) was identified close to nucleotide positions 2197–2727, covering approximately half of the AC1 region, including the AC4 ORF and the 3′ UTR. RDP4 indicated the event represents an Rbp of an isolate similar to ToLCNDV [Pk-06] (Acc. No. EF620534) found in *Luffa acutangula* in Pakistan and identified as putative minor parent into the background of ToLCNDV [BG-Jes-Svr-05] (Acc. No. AJ875157), found in tomato in Bangladesh, and identified as putative major parent. To the best of our knowledge, this is the first report of a ToLCNDV-ES recombinant isolate in the AC1-AC4 region in Italy.

## 1. Introduction

Tomato leaf curl New Delhi virus (ToLCNDV) is a bipartite member of the genus *Begomovirus,* family *Geminiviridae*, as it has two single-stranded DNA genome components denoted DNA-A and DNA-B (https://ictv.global/report/chapter/geminiviridae/geminiviridae/begomovirus (accessed on 13 January 2022)) [1]. The virus causes economically important damage in cultivated plant species of the *Solanaceae* and *Cucurbitaceae* families, and in at least 42 other dicotyledonous weeds, vegetable and ornamental species [2]. ToLCNDV is the most important tomato pathogen in India, where it attacks several other crops [3,4,5] but in recent years, it has extended this distribution and economic importance in the subtropical, tropical and temperate areas worldwide. ToLCNDV expansion was related to the high adaptive capacity of its main vector, the whitefly *Bemisia tabaci* [2,6,7], which was favored by mild temperatures and high humidity. Moreover, large areas of overlapping monocoltures grown in greenhouses and in open-fields of temperate regions contribute to the spread of the pathogen. These conditions led to ToLCNDV first detection in cucurbit crops of Spain in September 2012 [7,8,9,10,11], then in Tunisia [12], Italy [13,14,15,16], Morocco [17] and Algeria [18], with isolates closely related to those found in Spain. Both the DNA-A and DNA-B components from the ToLCNDV isolates found in the western Countries of the Mediterranean basin (WCMB) share nucleotide sequence similarities above 99%, which exceeds the strain demarcation boundary of 94% pairwise sequence identity for DNA-A proposed for begomoviruses [19]. Based on this threshold value, there is evidence that all the isolates found in WCMB belong to a unique strain denoted ToLCNDV-ES [10]. The isolates of the ToLCNDV-ES strain are better adapted to infect cucurbits rather than tomatoes and with minor field incidence and symptoms in tomato compared to cucurbit crops [10,11,20]. Such host adaptation and generation of severe epidemics may have been the outcome of a common adaptive recombination, which is extremely efficient in geminiviruses [21,22].

ToLCNDV is transmitted horizontally by *B. tabaci* in a circulative persistent manner and vertically in 25% seeds of Chayote squash (*Sechium edule*) [23] and in 60% of those of zucchini squash [24]. Disease symptoms in cucurbits are usually severe when the seedlings are infected during the trophic activity of insect vectors in that growth arrest occurs. The affected plants appear stunted and production is severely impaired. Leaves become malformed, curled and with thickened veins. They may also show chlorotic/necrotic vein banding, yellow speckles and mosaic of the leaf blade, whereas surface roughness and longitudinal cracking are frequently observed in fruits of cucurbits (Figure 1).

Beyond a certain similarity in the disease symptoms induced in various plant species, ToLCNDV is characterized by high genetic variability and many isolates have been reported, which account for the large number of crop plants affected and the severity of symptoms induced. The management of this disease needs constant monitoring because of the high propensity of ToLCNDV population to evolve through pseudorecombination and recombination at inter- and/or intra-species level [2,11,22,25].

The two circular single-stranded DNA molecules of the ToLCNDV genome are encapsidated within geminate particles [26]. DNA-A and DNA- B contain six (AV1, AV2, AC1, AC2, AC3, AC4) and two genes (BV1 and BC1), respectively. The DNA-A of some ToLCNDV isolates from cucumber, ridge gourd (*Luffa* spp.), pumpkin, tomato and zucchini squash codes also for a C5 protein with so far unknown function [27]. Genes in DNA-A and DNA-B are separated by an intergenic region (IR), encompassing a conserved 5′-TAATATT↓AC-3′ stem-loop with the breaking and joining site (arrow) for rolling-circle replication [28,29]. ToLCNDV may be also associated with betasatellites, which contain a single gene encoding a pathogenicity determinant that exacerbates disease symptoms. Both DNA-B and betasatellites depend on the replicase coded by DNA-A for their replication [30].

The aim of this study was to determine the primary structure of DNA-A and DNA-B of the ToLCNDV-Le isolate, which was obtained from a single plant of zucchini squash grown in a badly affected crop of a group of greenhouses plant in the Lecce Province (southern Italy). Detection, identification and biological characterization of the ToLCNDV-Le isolate has been recently reported (Figure 1) [31].

## 2. Results

### 2.1. Sequence Analysis

Due to the poor condition of the naturally infected zucchini plant, which yielded nucleic acid preparations of poor purity, the complete nucleotide sequence of ToLCNDV-Le genome was determined on nucleic acid extracts prepared from one plant of *L. siceraria* to which the natural virus infection was transferred by a single rub-inoculation with sap of the naturally infected zucchini plant. *L. siceraria* was chosen as it ranked as a susceptible host in the biological characterization of ToLCNDV-Le [31].

After parsing, sequences with poor quality scores were removed from raw reads, obtaining 47,902,003 reads of 150 bp in length. Paired reads were mapped to the *Lagenaria siceraria* reference genome (Acc. No GCA_002890555.2). The unmapped reads (69.5%) generated two main contigs using de novo assembly tool of CLC (CLC GENOMICS WORKBENCH v.22.0) and the ToLCNDV isolate 661 Almeria (Acc. No. KF749223.1 for DNA A and Acc. No. KF749226.1 for DNA B) as reference genome. Like other Old World (OW) bipartite begomoviruses, the complete ToLCNDV-Le genome encompasses DNA-A and DNA-B of 2738 and 2683 nt in length, deposited in the EMBL/GeneBank under the Acc. No. OQ262954 and OQ262955, respectively. In silico translation of DNA-A and DNA-B sequences revealed an organization similar to that from other OW bipartite begomoviruses, consisting of seven and two putative ORFs coded by DNA-A and DNA-B, respectively (Table 1, Appendix A), plus an IR region in common between the two DNA molecules, containing the conserved nonanucleotide sequence 5′-TAATATT↓AC-3′. No betasatellite DNA was detected in ToLCNDV-Le samples of zucchini squash with natural infection nor in three rub-inoculated plants of zucchini squash cv. President and *L. siceraria* as evaluated by PCR using the universal betasatellite primers [32]. In order to validate the whole genome sequencing data analysis (WGS), PCR reaction verification was performed yielding the expected size fragment of 1048 bp, comprising the Coat protein AV1 gene, from the genomic DNA of infected zucchini plants. Double Sanger sequencing of the PCR products with SP6 and T7 universal primers was performed revealing 99.62% identity (data not shown) with the complete DNA-A genome assembled from the Illumina reads, thus validating the accuracy of the sequences within the assembled virus genomes. The ToLCNDV-Le Coat protein/AV1 gene sequence was deposited in GenBank under the accession number ON783713.

### 2.2. Recombination Analysis

Genetic analysis of the recently emerged ToLCNDV isolates in Spain that efficiently infect cucurbit species, putatively diverged from isolates reported from other parts of the world by recombination [10,11]. Therefore, we examined the ToLCNDV-Le sequence for any evidence of recombination by comparing the DNA-A of the ToLCNDV-Le with that of 119 ToLCNDV isolates available in the NCBI-GenBank and International Committee on Taxonomy of Viruses (ICTV) databases as at July 2022, selected on the basis of the following criteria: (i) isolates from cucurbits, Solanaceae and weeds; (ii) isolates from WCMB; and (iii) isolates in the list of the ToLCNDV exemplar species designated by the ICTV (https://ictv.global/report/chapter/geminiviridae/geminiviridae/begomovirus (accessed on 13 January 2023)) [1]. Six out of the nine different algorithms implemented in the RDP4 package identified one putative Rbp at nt position 2197–2727, which covered approximately 72% of AC1 ORF, including the complete AC4 ORF and part of the 3′ UTR (Figure 2 and Table 2).

The Rbp was identified by RDP; MaxChi; Chimaera, SiScan, 3 Seq and LARD (Table 2).

The event at nt positions 2197–2727 is likely to represent an Rbp of a virus similar to ToLCNDV [PK-05] (Acc. No. EF620534) isolated from *Luffa acutangula* in Pakistan and identified as minor parent. The same analysis identified the ToLCNDV [BG-Jes-Svr-05] (Acc. No. AJ875157), found in tomato samples showing severe leaf curling and stunting collected at Jessore in Bangladesh [6], as putative major parent by all the abovementioned six algorithms in the RDP4 package. The recombinant structure exhibited by ToLCNDV-Le DNA-A, covering part of AC1 ORF and the entire AC4 ORF, is similar to the recombinant sequences described by Fortes et al. [11] as events 3 and 5 found in ToLCNDV-[IN-Jun-TC309-11] (Acc. No. KF551576) isolated from tomato in India with an Rbp at nt positions 1397–2301 and ToLCNDV-[IN-Har-Lc-07] (Acc. No. FN645905) isolated from *L. siceraria* in India with an Rbp at nt positions 2063–2661, respectively.

### 2.3. Phylogenetic Affinities of ToLCNDV-Le Complete Genome

Phylogenetic trees were constructed using full-length DNA-A and DNA-B sequences as well as sequences of AC1 and AC4 ORFs that were presumably involved in the 2197–2727 Rbp detected in the DNA-A of the ToLCNDV-Le isolate (Figure 2). The phylogenetic analysis encompassed all the sequences used for the analysis with the RDP4 package, including the isolates indicated as major and minor parent for the putative 2197–2727 Rbp by the RDP4 analysis, together with the isolates ToLCNDV-[IN-Jun-TC309-11] (Acc. No. KF551576) and ToLCNDV-[IN-Har-Lc-07] (Acc. No. FN645905) carrying the Rbp events denoted 3 and 5, respectively, by Fortes et al. [11]. These two events cover the AC1 and AC4 ORF regions similarly to the 2197–2727nt Rbp found in the ToLCNDV-Le isolate. According to these criteria, a total of 119 full-length DNA-A and 64 full-length DNA-B sequences were retrieved from public databases as at July 2022. Tomato mosaic Havana virus-[Quivican] DNA-A (Acc. No. Y14874.1) and DNA-B (Acc. No. Y14875.1) were used as outgroups.

The DNA-A sequence of ToLCNDV-Le clustered with isolates of the ES strain from WCMB with greater than 99% nucleotide sequence identity (Figure 3 and Appendix A) that seven of such isolates were from Spain: ToLCNDV [ES-MU-8.1-Sq-12] (Acc. No. KF749224) from *Cucurbita pepo*; ToLCNDV-[ES-Alm-661-Sq-13] (Acc. No. KF749223) from *C. pepo*; ToLCNDV [ES-MU-11.1-Sq-12] (Acc. No. KF749225) from *C. pepo*; ToLCNDV [ES-Alm-TomA5-14] (Acc. No. KT175406) from *Solanum lycopersicum*; ToLCNDV [ES-Alm-Zucchini-13] (Acc. No. KF891468) from *C. pepo*; ToLCNDV [ES-Alm-Cuc-16] (Acc. No. LC596380) from *Cucumis sativus*; ToLCNDV [ES-Alm-TomA4-14] (Acc. No. KM977733) from *S. lycopersicum*. All the Spanish isolates putatively harbour the recombination event 4 reported by Fortes et al. [11], together with one from Morocco, the ToLCNDV isolate Agadir (Acc. No. MG098230), identified in *C. pepo.* Significantly separated branches of the same node included five isolates from India ToLCNDV [IN-RG1-RG5-13] (in the phylogenetic tree, from the Acc. No. KT426903 to the Acc. No. KT426907) identified in *Luffa acutangula*, and one isolate from *Sechium edule* ToLCNDV [IN-TN-TDK-CHOU2-14] (Acc. No. KP191047), which exhibited 91−93% nucleotide similarity with the ES isolates (Figure 3 and Appendix A) and harbour recombination event 4 according to Fortes et al. [10]. ToLCNDV virus 5 [BD-cuc-06] (Acc. No. EF450316) from *C. sativus*, which is an exemplar species recognized by ICTV, and two Pakistan isolates ToLCNDV [PK-Sn-PT10-04] PK (Acc. No. DQ116883) and ToLCNDV PK isolate Rahim Yar Khan 2 (Acc. No. DQ116885), both from *S. lycopersicum*, formed separated branches of the same node.

Phylogenetic analysis of the isolates for which the full-length DNA-B sequence was available (Figure 4), revealed a topological group of isolates of ToLCNDV-ES strain including the ToLCNDV-Le isolate with 99−97% nucleotide similarities (Figure 4 and Appendix A), robustly separated from all other ToLCNDV isolates.

Phylogenetic relationships of AC1 nucleotide sequences (Appendix A) are substantially similar to those of full DNA-A as ToLCNDV-Le clustered with isolates of the ES strain with a nucleotide sequence identity of about 99% (Appendix A and Appendix A). Amino acid sequence analysis and folding of AC1 protein translated from ToLCNDV-Le recombinant sequence showed only two mutations in positions 102 and 212 of the AC1 protein sequence compared to the AC1 translated protein of ToLCNDV [ES-MU-11.1-Sq-12] SPAIN (Figure 5).

The substitutions do not affect the folding of AC1 protein as observed by the structure prediction analysis putatively allowing the recognition of the DNA-binding domain of the replication initiation protein from the geminivirus like in Tomato Yellow leaf curl Sardinia virus in the first 121 amino acids of the sequence [37] (Figure 5). Similar relationships were also observed among AC4 sequences (Appendix A), but ToLDNDV-Le sharing 100% nucleotide similarity with eight out of the nine isolates of the ES strain (Appendix A and Appendix A) and the isolate ToLCNDV [ES-MU-11.1-Sq-12] SPAIN (Acc. No. KF749225) from *C. pepo* with which the virus shared only 99,40% nucleotide sequence similarity (Appendix A and Appendix A).

## 3. Discussion

In this study, results from the analysis based on a pairwise comparison of a number of full-length DNA-A sequences retrieved from public databases and ICTV demarcation criteria for begomoviruses demonstrated that the ToLCNDV-Le isolate belongs to the ES strain and is significantly differentiated from isolates and recombinant strains reported from the Indian subcontinent and Asian Countries. Thus, the ToLCNDV-Le isolate probably emerged by the putative recombination event 4 in the AC2-AC3 coding region, which has been recorded in ToLCNDV [ES-Alm-661-Sq-13] (Acc. No. KF749223); ToLCNDV [IN-TN-TDK-CHOU2-14] (Acc. No. KP191047) and ToLCNDV [IN-RG1-13] (Acc. No. KT426903) as suggested by Fortes et al. [11]. However, the analysis of the whole nucleotide sequence of the ToLCNDV-Le isolate revealed another putative Rbp in DNA-A. This analysis highlighted a significant hot-spot for recombination with *p* values < 0.05 across the center of AC1, the complete sequence of the AC4 ORF and about 144 nt of the stretch upstream of the IR region. The position of the Rbp found in this study fits well with the recombination hot-spots identified by Lefeuvre et al. [38], who analyzed the sequences of begomoviruses with multiple computational tools. In that study, the AC1 center was identified as hot-spot for recombination. However, since the AC4 ORF is embedded in the AC1 sequence, the Rbp across the AC1 spans the AC4 ORF making the AC4 region bounded by two recombination hot-spots, i.e., part of the AC1 and the 3′-UTR [34]. The product of the AC4 ORF is a pathogenicity determinant but it is also involved in the suppression of RNA silencing [39]. This protein function is also supported by the 15KDa protein translated from the AC2 ORF, which is a pathogenicity determinant and also a suppressor of RNA silencing [40]. The AC2 ORF is apparently not involved in the Rbp reported here, although AC2 and AC3 have been indicated as hot spots for recombination [34] and reported as event 4 by Fortes et al. [11] from which the ES strain emerged. Indeed, event 4 has been detected in ToLCNDV [ES-Alm-661-Sq-13] (Acc. No. KF749223) which is an isolate of the ToLCNDV-ES strain [11]. Thus, it is tempting to speculate that the ToLCNDV-Le isolate emerged by the recombination in the AC2-AC3 region as the other ES strains reported from WCMB, and that it probably underwent a second recombination event in the AC1-AC4 coding region or this Rbp preceded the one in the AC2-AC3 coding region. This hypothesis, obviously awaits further studies.

Another point that requires further investigation is whether AC1-AC4 Rbp altered the characteristics of AC4 as a suppressor of RNA silencing. However, this might not be the case, since Lefeuvre et al. [41] demonstrated that recombination events tend to be poorly disruptive in regard to the folding of protein translated from recombinant sequences. Our results with structure-based sequence alignment of the AC1 and AC4 protein from ToLCNDV-Le (Acc. No. OQ262954) and ToLCNDV_ES (Acc. No. KF749225.1) support this hypothesis.

In conclusion, results from this study suggest that ToLCNDV-Le is an isolate of the ToLCNDV-ES for which the name and descriptors ToLCNDV-ES [IT-Zu-Le-18] are proposed.

## 4. Materials and Methods

### 4.1. Whole Genome Sequencing (WGS) and Assembly

Natural infection of ToLCNDV-Le found in a badly affected zucchini plant grown in the Province of Lecce (southern Italy) was identified by dot-blot hybridization with a Digoxigenin-labeled (DIG) DNA [31]. Inoculum from this plant was used to obtain the ToLCNDV-Le isolate, which had been characterized biologically [31]. ToLCNDV-Le was transferred to and maintained in a number of herbaceous hosts grown in laboratory greenhouses. *Lagenaria siceraria* was a very susceptible host [31]; thus, ToLCNDV-Le genome sequence was obtained after a single rub-inoculation on *L. siceraria* of the naturally infected zucchini plant. *L. siceraria* was used as source for nucleic acid preparation suitable for whole genome sequencing (WGS).

The full-length sequence of ToLCNDV-Le DNA-A and DNA-B was determined from total nucleic acids preparations obtained from 100 mg of ToLCNDV-Le infected leaves of *L. siceraria,* following the method of Dalmay et al. [42]. Quality and quantity of DNA were checked by 1% (*w*/*v*) agarose gel electrophoresis in 1× TBE buffer (90 mM tris-base, 90 mM boric acid, 2 mM Na_2_EDTA), and by spectra absorption analysis (NanoDrop 2000 Spectrophotometer, Thermo Scientific Inc., Wilmington, DE, USA). Quantification of DNA samples for library preparation was assessed using the Qubit DNA broad rangeAssay (Invitrogen, Paisley, UK).

The Illumina NovaSeq 2 × 150 bp was used for the sequencing of DNA libraries with a 100% depth of coverage. After parsing, sequences with poor quality scores were removed from raw reads with the FastQC tool (www.bioinformatics.babraham.ac.uk/projects/fastqc/ (accessed on 5 July 2022)) [43] and low-quality bases at the 3′ ends of reads were trimmed using a quality threshold filter of 20 on Galaxy platform [44,45]. High-quality reads were mapped to the *L. siceraria* genome sequence (Acc. No. GCA_002890555.2; genome of 297.9 Mb in size divided in 203 contigs) used as reference and against chloroplast and mitochondrium of *L. siceraria* assembled by get.organelle program (v. 1.7.6.1) using CLC (CLC GENOMICS WORKBENCH v.22.0). The unmapped reads were used to assemble the ToLCNDV-Le genome with the de novo assembly tool of CLC, with default settings. The assembled DNA sequences (contigs) were aligned on ToLCNDV reference genome (ToLCNDV isolate 661 Almeria Acc. No. KF749223.1 for DNA A and Acc. No. KF749226.1 for DNA B) and used to fill gaps between contigs.

Open reading frames (ORFs) in the assembled ToLCNDV genome and protein prediction were determined using the SeqBuilder Pro tool (Lasergene v.15.0.1; DNASTAR) and the ORF Finder program run on-line (https://www.ncbi.nlm.nih.gov/orffinder/ (accessed on 20 July 2022)) [33].

### 4.2. Betasatellite Detection in ToLCNDV-Le Infected Plants

Total DNA extracts were from leaves of the badly affected zucchini plant with natural ToLCNDV-Le infection and from leaves of three samples of zucchini squash cv. President and *L. siceraria* rub-inoculated with the sap obtained from naturally infected tissues of zucchini plant, as described by Mastrochirico et al. [31], were also tested for the presence of betasatellite DNA. One hundred ng of total DNA preparations obtained following the CTAB method [46] were used as template for PCR amplification with the universal primer pair for DNA betasatellites 5′-GGTACCACTACGCTACGCAGCAGCC-3′ and 5′-GGTACCTACCCTCCCAGGGGTACAC-3′ [32]. PCR reactions were carried out using the following condition: 35 cycles of melting at 94 °C for 1 min; annealing at 50 °C for 1 min; extension for 1.5 min at 72 °C. PCR products were visualized on 1.2% agarose gel electrophoresis in 1× TBE buffer and gel-red staining.

### 4.3. Sequence Validation through Polymerase Chain Reaction (PCR) and Sanger Sequencing

Total DNA was extracted from systemically infected *L. siceraria* leaves following the CTAB method [46] and 100 ng were used to perform PCR amplification of the viral AV1 gene. PCR reactions were carried out using the following cycle: 3 min denaturation at 94 °C followed by 35 cycles of 30 s denaturation at 94 °C, 30 s annealing at 55 °C and 20 s synthesis at 72 °C with the final elongation step for 5 min at 72 °C. Electrophoresis check in 1.2% agarose gel in TBE buffer (90 mM Tris, 90 mM boric acid, 1 mM EDTA) and gel-red staining was performed to verify primer specificity. Primer pairs sequences used to amplify the ToLCNDV-Le-AV1 DNA fragment of 1149 bp were Forward-CP1 5′CTCCAAGAGATTGAGAAGTCC3′ and reverse-CP2 5′TCTGGACGGGCTTACGCCCT3′ as reported by Panno et al., 2019 [47]. The PCR product was ethanol-purified and quantified using a Nanodrop 2000 spectrophotometer (Thermo Fisher Scientific, Waltham, MA, USA) to determine purity levels, then was ligated to pGEMT-easy vector (Promega) (Promega, Charbonnières-Les Bains, France), cloned in Escherichia. coli and Sanger-sequenced (Genewiz from Azenta Life Sciences) using T7 and SP6 universal primers for the double direction validation of the ToLCNDV-Le CP sequence. Comparisons were made with the complete DNA-A ToLCNDV-Le sequences assembled from Illumina reads.

### 4.4. Phylogeny Relationships and Evidence of Recombination

Bioedit [48] and pairwise multiple sequence alignments were produced using the MUSCLE algorithm implemented in MEGA XI [49] employing DNA-A ToLCNDV-Le with 119 DNA-A and 64 DNA-B ToLCNDV isolates retrieved from GenBank as at July 2022 (Acc. No. indicated in parenthesis).

In order to detect potential recombination events among the nucleotide sequences of the ToLCNDV-Le isolate with 119 isolates selected for DNA-A, the RDP program suite version 4 (RDP v.4.13) [50], i.e., the algorithms RDP, GENECONV, BOOTSCAN, MAXCHI, CHIMAERA, SISCAN, 3SEQ, LARD and PHYLPRO [51] were used. Automated analysis was carried out using the default RDP4 settings and only potential recombination events independently identified by two or more detection methods were taken into consideration and deemed statistically significant with a Bonferroni-corrected *p*-value ≤ 0.05.

Phylogeny analyses were performed for ToLCNDV-Le DNA-A and DNA-B full-length ORF nucleotide sequences within and between isolates selected from the Mediterranean and Asian areas and retrieved from GenBank (Acc. No. indicated in parenthesis). The evolutionary history was inferred using the Neighbor–Joining method [34] in MEGA XI. Statistical support for the branches was evaluated using a bootstrap analysis with 1000 replicates while the full sequence of tomato mosaic Havana virus (ToMHaV) Quivican isolate from CUBA was selected as outgroup for DNA-A (Acc. No. Y14874.1) and DNA-B (Acc. No. Y14875.1). The percentage of replicate trees in which the associated taxa clustered together in the bootstrap test are shown next to the branches and only bootstrap values >50% are indicated in the nodes. [52]. The tree is drawn to scale, with branch lengths in the same units as those of the evolutionary distances used to infer the phylogenetic tree. The evolutionary distances were computed using the Tamura 3-parameter method [53] and are in the units of the number of base substitutions per site.

### 4.5. Protein Structural Analyses

The Clustal Omega tool (https://www.ebi.ac.uk/Tools/msa/clustalo/ (accessed on 3 April 2023)) [35] was used to analyze the multiple sequence alignment. Differences of amino acid properties based on the chemical composition of residual group was visualized using PSIPRED Workbench (http://bioinf.cs.ucl.ac.uk/psipred/ (accessed on 4 April 2023)) [36]. The three-dimensional structure predicted for the first 121 amino acids of protein sequences was based on PDB ID 1L2M protein structure. β-strands and α-helices motif analysis was obtained using the PSIPRED Workbench of the complete AC1 sequence.

## 5. Conclusions

Taken together, our results support that a hitherto identified recombinant isolate in the AC1-AC4 coding region of ToLCNDV-ES strain was present in one plant of zucchini squash in the Apulia Region (southern Italy). So far we do not know to what extent it is present in the region or if it infects other cucurbit and non-cucurbit species. Concerns, thus, exist associated with its potential progressive spread and threat to additional vegetable crops; in particular, if mixed infections occur with other viruses of vegetable crops. In the framework of a Regional Program for the protection of biodiversity, results from an on-going survey is aimed to address these aspects.

## Figures and Tables

**Figure 1 plants-12-02399-f001:**
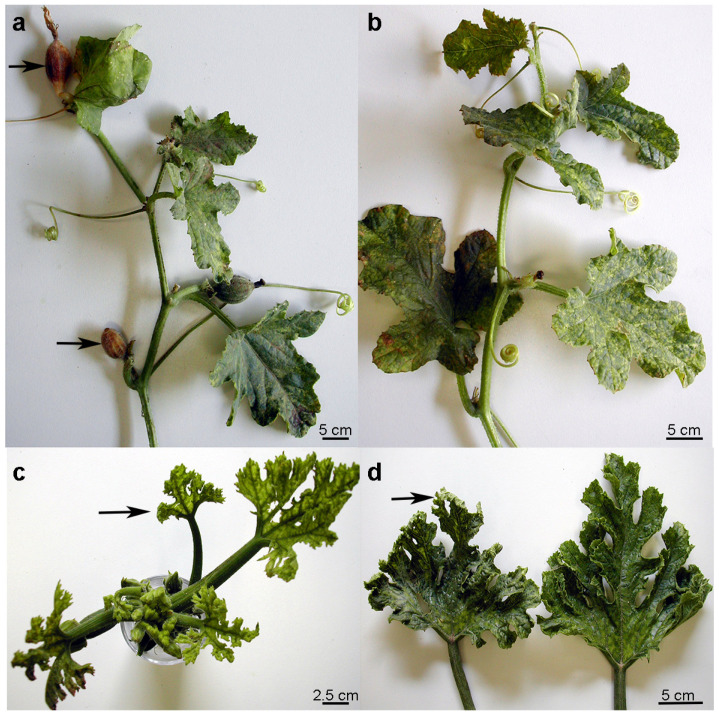
Severe disease symptoms caused by Tomato leaf curl New Delhi virus in early-infected plants of melon (**a**,**b**) and zucchini squash (**c**,**d**). In panel (**a**), the melon plant with necrotic fruits set indicated by arrows. In panels (**c**,**d**), zucchini leaves showing chlorotic/necrotic mosaic, thickened leaf lamina and upside-curled margins indicated by arrows. Scale bars indicate plant material size.

**Figure 2 plants-12-02399-f002:**
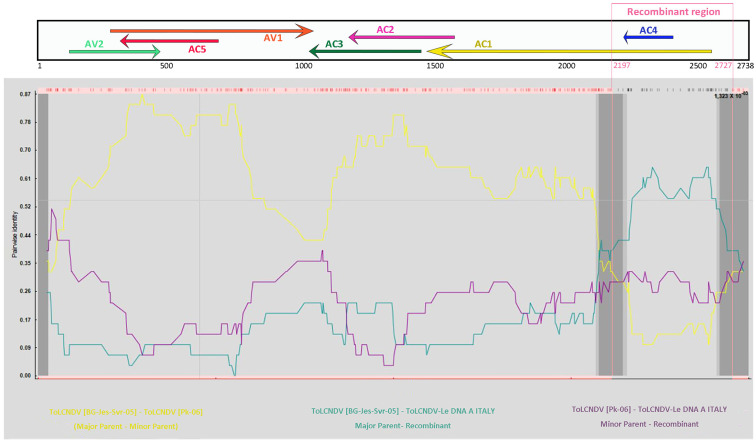
Evidence of the putative Rbp (highlighted in pink in the schematic representation of the ToLCNDV-Le genome) at nt positions 2197−2727 of the ToLCNDV-Le DNA-A detected by Recombination detection program (RDP, v4.13) algorithms (*p* = 1.323 × 10^−3^). The sequences of ToLCNDV [BG-Jes-Svr-05] (Acc. No. AJ875157) and ToLCNDV [PK-05] (Acc. No. EF620534) represent putative major and minor parents, respectively.

**Figure 3 plants-12-02399-f003:**
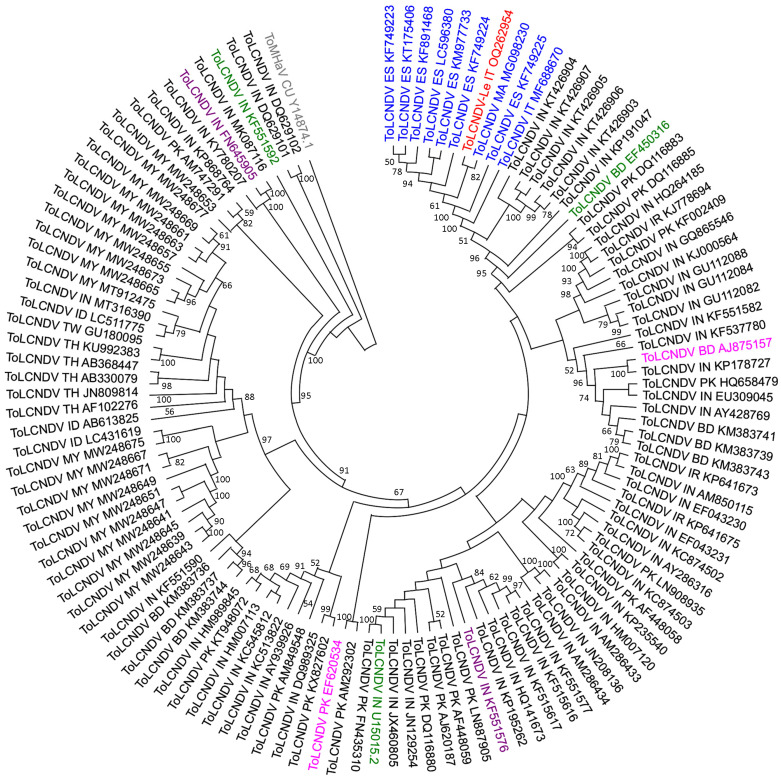
Phylogenetic relationships of full-length DNA-A sequence of ToLCNDV-Le (in red) with that of 119 sequences selected from ToLCNDV public databases as at July 2022, which include: (i) isolates from Countries in the western Mediterranean Basin (in blue); (ii) isolates in the list of the ToLCNDV exemplar species designated by the ICTV (in green); (iii) the two isolates identified by the RDP4 package as putative major and minor parents (in pink) of the Rpb in the DNA-A of ToLCNDV-Le; (iv) isolates ToLCNDV-[IN-Jun-TC309-11] (Acc. No. KF551576) and ToLCNDV-[IN-Har-Lc-07] (Acc. No. FN645905) (in violet) carrying the Rbp events denoted 3 and 5, respectively, by Fortes et al. [11]; (v) sequence of the DNA-A of isolate (CU-Qui) of the bipartite begomovirus species tomato mosaic Havana virus (ToMHaV) (Acc. No. Y14874.1) as outgroup (in grey). The abbreviations used for the country of origin for the sequences are as follows: BD, Bangladesh; CU, Cuba; ES, Spain; ID, Indonesia; IN, India; IR, Iran; IT, Italy; MA, Morocco; MY, Malaysia; PK, Pakistan; TH, Thailand; TW, Taiwan. The GenBank accession number is indicated for each isolate included in the analysis. The tree was constructed using the Neighbor–Joining method [34] in MEGA11 (Molecular Evolutionary Genetics Analysis v.11.0.13). Statistical support for the branches was evaluated using a bootstrap analysis with 1000 replicates.

**Figure 4 plants-12-02399-f004:**
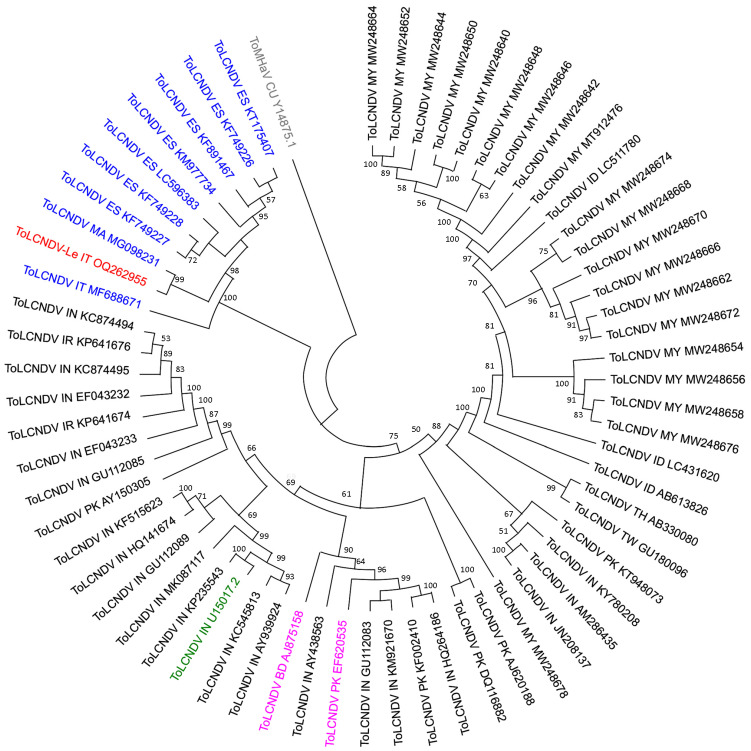
Phylogenetic relationships of full-length DNA-B sequence of ToLCNDV-Le (in red) with that of 64 sequences selected from ToLCNDV public databases as at July 2022, which include: (i) isolates from Countries in the western Mediterranean Basin (in blue); (ii) isolates in the list of the ToLCNDV exemplar species designated by the ICTV (in green); (iii) the two isolates identified by the RDP4 package as putative major and minor parents (in pink) of the Rpb in the DNA-A of ToLCNDV-Le; (iv) the sequence of the DNA-B of isolate (CU-Qui) of the bipartite begomovirus species tomato mosaic Havana virus (ToMHaV) (Y14875.1) as outgroup (in grey). The abbreviations used for the country of origin for the sequences are as follows: BD, Bangladesh; CU, Cuba; ES, Spain; ID, Indonesia; IN, India; IR, Iran; IT, Italy; MA, Morocco; MY, Malaysia; PK, Pakistan; TH, Thailand; TW, Taiwan. The GenBank accession number is indicated for each isolate included in the analysis. The tree was constructed using the Neighbor–Joining method [34] in MEGA11 (Molecular Evolutionary Genetics Analysis v.11.0.13). Statistical support for the branches was evaluated using a bootstrap analysis with 1000 replicates.

**Figure 5 plants-12-02399-f005:**
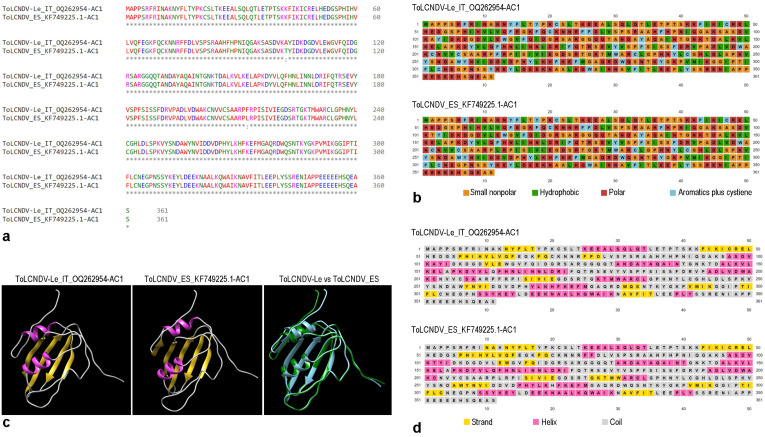
Structure-based sequence alignment of the AC1 protein from ToLCNDV-Le (Acc. No. OQ262954) and ToLCNDV_ES (Acc. No. KF749225.1). In panel (**a**), conserved residue in amino acids sequence alignments are highlighted by asterisks (*), whereas a colon (:) indicates conservation between groups of strongly similar properties. The Clustal Omega tool (https://www.ebi.ac.uk/Tools/msa/clustalo/ (accessed on 3 April 2023)) [35] was used to analyze the multiple sequence alignment. Panel (**b**) showed differences of amino acid properties based on the chemical composition of residual group visualized using PSIPRED Workbench (http://bioinf.cs.ucl.ac.uk/psipred/ (accessed on 4 April 2023)) [36]. Panel (**c**) displayed the three-dimensional structure predicted for the first 121 amino acid of protein sequences based on PDB ID 1L2M protein structure. Secondary structure elements are indicated by cylinders (α-helices, in pink) and arrows (β-strands, in yellow) above and below the coil sequence (in grey). β-strands and α-helices motif analysis using the PSIPRED Workbench of the complete AC1 sequence is reported in panel (**d**).

**Table 1 plants-12-02399-t001:** Putative ORFs identified in DNA-A and DNA-B of ToLCNDV-Le genome by SeqBuilder Pro (Lasergene v.15.0.1; DNASTAR) and the ORF Finder programs (https://www.ncbi.nlm.nih.gov/orffinder/ (accessed on 20 July 2022)) [33].

ORF *	Protein Function	Strand	Start	Stop	ORF Length (nt)	ORF Coding Capacity
AV1	Coat	V	279	1049	771	256
AV2	Precoat	V	119	457	339	112
AC1	Replication initiation	C	2583	1498	1086	361
AC2	Transcription activator	C	1595	1191	405	134
AC3	Replication enhancer	C	1456	1046	411	136
AC4	Pathogenesis-related	C	2426	2250	117	58
AC5	Unknown	C	671	309	363	120
BV1	Nuclear shuttle	V	430	1236	807	268
BC1	Movement	C	2135	1290	846	281

* Putative ORFs preceded by DNA component designation (A or B); V = virion-sense strand; C = complementary-sense strand.

**Table 2 plants-12-02399-t002:** Identification of a putative 530 nt-long Rbp in the ToLCNDV-Le DNA-A genome highlighted by RDP4 software between nt 2197 and 2727. The ToLCNDV sequences with Acc. No. EF620534 and AJ875157 are likely to represent parental sequences identified by the RDP4 software.

Recombinant ^1^	Potential Parents ^2^		Recombination Breakpoint	Average *p* Values in Detecting Algorithms ^3^					
	Minor	Major		R	M	C	S	T	L
ToLCNDV-Le DNA-A	ToLCNDV [PK-05] (EF620534)	ToLCNDV [BG-Jes-Svr-05] (AJ875157)	2197–2727	1.323 × 10^−3^	1.724 × 10^−2^	7.821 × 10^−3^	4.982 × 10^−12^	8.523 × 10^−6^	7.144 × 10^−5^

^1^ ToLCNDV-Le DNA-A: Tomato leaf curl New Delhi virus-Le DNA-A full genome sequence. ^2^ ToLCNDV Acc. No. EF620534 and AJ875157. ^3^ R: RDP, M: MaxChi, C: Chimaera, S: SiScan, T: 3Seq, L: Lard.

## Data Availability

ToLCNDV-Le sequences data are available in NCBI under the Acc. No. OQ262954 for DNA-A, Acc. No. OQ262955 for DNA-B and Acc. No. ON783713 for Coat protein/AV1 gene.

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
