# Peer review of "Molecular Characterization of a Recombinant Isolate of Tomato Leaf Curl New Delhi Virus Associated with Severe Outbreaks in Zucchini Squash in Southern Italy"

_plants, 2023, doi:10.3390/plants12132399_

Round 1
Reviewer 1 Report
Plants-2427632
Mastrochirico et al presented this manuscript entitled “Molecular characterization of a recombinant isolate of tomato leaf curl New Delhi virus associated with severe outbreaks in zucchini squash in southern Italy “for publication consideration in Plants.
This manuscript reported on molecular characterization of a tomato leaf curl New Delhi virus (ToLCNDV) isolate, denoted ToLCNDV-‐‑Le, which cause severe problems with protected crops of zucchini squash grown in the Province of Lecce, Apulia in southern Italy.
Genomic studies and phylogenetic analysis and other methods were performed in this report. A sounds conclusion was presented. It is first report of a ToLCNDV-‐‑ES recombinant isolate in the AC1-‐‑AC4 region in Italy.
The manuscript is well organized and presented. Background introduction is sufficient, M&M is clearly descripted, results and discussion sections are well written.
The scientific soundness of this manuscript is acceptable. It meets the aims and scope of Plants journal.
This study is important and interesting. I believe this manuscript should be accepted for publication with Plants journal after some revision. My common regarding the manuscript list below:
1. In Fig 1. Symptoms caused by Tomato leaf curl New Delhi virus. No scale bar used here and no mention about what plant materials presented and clear indication like arrow to point out symptom.
2. With discussion section, what kind of application could be driven out from this study is lacking, authors may want to add some points and perspectives here.
3. Should there be a conclusion section?
Author Response
Dear Editor Gary Cao,
thank you for your message.
The Authors thank the Reviewers for all the suggestions to improve the Manuscript (ID plants-2427632) entitled “Molecular characterization of a recombinant isolate of tomato leaf curl New Delhi virus associated with severe outbreaks in zucchini squash in southern Italy” by Mariarosaria Mastrochirico, Roberta Spanò, Rita Milvia De Miccolis Angelini, Tiziana Mascia.
Please find below in red our answers to the reviewers' comments:
Reviewer 1
Mastrochirico et al presented this manuscript entitled “Molecular characterization of a recombinant isolate of tomato leaf curl New Delhi virus associated with severe outbreaks in zucchini squash in southern Italy “for publication consideration in Plants.
This manuscript reported on molecular characterization of a tomato leaf curl New Delhi virus (ToLCNDV) isolate, denoted ToLCNDV‐Le, which cause severe problems with protected crops of zucchini squash grown in the Province of Lecce, Apulia in southern Italy.
Genomic studies and phylogenetic analysis and other methods were performed in this report. A sounds conclusion was presented. It is first report of a ToLCNDV-ES recombinant isolate in the AC1-AC4 region in Italy.
The manuscript is well organized and presented. Background introduction is sufficient, M&M is clearly descripted, results and discussion sections are well written.
The scientific soundness of this manuscript is acceptable. It meets the aims and scope of Plants journal.
This study is important and interesting. I believe this manuscript should be accepted for publication with Plants journal after some revision.
R. We thank the Reviewer for the positive evaluation and for all the suggestions to improve the Manuscript (ID plants-2427632).
My common regarding the manuscript list below:
- In Fig 1. Symptoms caused by Tomato leaf curl New Delhi virus. No scale bar used here and no mention about what plant materials presented and clear indication like arrow to point out symptom.
R: We revised the Figure 1 supplementing a more detailed description of disease symptoms. We added arrows to indicate the symptoms described in the figure legend and scale bars to indicate plant material size.
2. With discussion section, what kind of application could be driven out from this study is lacking, authors may want to add some points and perspectives here.
R. We included comments on the outcomes of this study and future exploitation of the results in the conclusion section
3. Should there be a conclusion section?
R. We added the Conclusion section, as requested.
We hope that all the explanations reported above are good enough to proceed with the positive evaluation of our manuscript for publication in Plants.
With many thanks for your attention and time.
Please address all correspondence concerning this manuscript to me.
Best regards,
Dr. Tiziana Mascia (PhD)
Reviewer 2 Report
The main aim of the paper is the molecular characterization of a tomato leaf curl New Delhi virus (ToLCNDV) isolate, denoted ToLCNDV--‐‑Le.
The paper shows an interesting information about the isolate, but the number of samples is unknown and if just one sample has been analysed the study has a serious limitation. More samples should be included to give a clue about the extent of this isolate. And more information is needed in some parts of the paper.
Specific comments
Line 86. Functions of those genes should be indicated.
Line 99. This paragraph gives results and conclusions. Here should the objective of the paper, not the results.
Line 143. In material and methods 120 samples are reported. Which is the total number?
Line 307. This part is not related to the rest of the paper. Why is it included in? More information in the previous parts of the paper is needed to include this conclusion.
Line 329. More information about the origin of the samples should be given. Just one sample? No other samples isolated from other crops or fields? More samples should be included in the study.
Samples were from Lagenaria? Why not from cultivated species of Cucurbita?
Line 354. How many plants were used from Zucchini? All from the same field? Different isolates? The three samples from Lagenaria are from different fields? Different isolates?
Author Response
Dear Editor Gary Cao,
thank you for your message.
The Authors thank the Reviewers for all the suggestions to improve the Manuscript (ID plants-2427632) entitled “Molecular characterization of a recombinant isolate of tomato leaf curl New Delhi virus associated with severe outbreaks in zucchini squash in southern Italy” by Mariarosaria Mastrochirico, Roberta Spanò, Rita Milvia De Miccolis Angelini, Tiziana Mascia.
Please find below in red our answers to the reviewers' comments:
Reviewer 2
The main aim of the paper is the molecular characterization of a tomato leaf curl New Delhi virus (ToLCNDV) isolate, denoted ToLCNDV-Le.
The paper shows an interesting information about the isolate, but the number of samples is unknown and if just one sample has been analysed the study has a serious limitation. More samples should be included to give a clue about the extent of this isolate. And more information is needed in some parts of the paper.
R. We thank the Reviewer for all the suggestions to improve the Manuscript (ID plants-2427632). The aim of this study was to determine the primary structure of the ToLCNDV-Le isolate, which was obtained from a single zucchini plant grown in a badly affected zucchini crops of a group of greenhouses plant in the Lecce Province (southern Italy). Thus, in our opinion, an extended survey about the extent of this isolate on other zucchini plants and other plant species was out of the scope of this study and, in turn, we do not see why the characterization of a single isolate limits seriously the validity of this study. We added a conclusion section upon suggestion of Reviewer 1 and there we commented the outcome of this study as well as future work concerning the occurrence of this ToLCNDV-ES isolate recombinant in the AC1-AC4 coding region in other cucurbits and vegetable species.
Specific comments
Line 86. Functions of those genes should be indicated.
R. Functions of ToLCNDV genes are reported in Table 1. To avoid redundancy, we preferred to list these functions only in Table 1, which refers to ToLCNDV-Le genome.
Line 99. This paragraph gives results and conclusions. Here should the objective of the paper, not the results.
R. Generally, the introduction ends with the anticipation of more relevant results. In addition we shortened the paragraph by deleting lines 104-108.
Line 143. In material and methods 120 samples are reported. Which is the total number?
R. In the Section 4.4 Phylogeny relationships and evidence of recombination, we examined the ToLCNDV-Le sequence using the full-length DNA-A sequences of 119 ToLCNDV isolates and full-length DNA-B sequences of 64 ToLCNDV isolates available in the NCBI-GenBank and International Committee on Taxonomy of Viruses (ICTV) databases as July 2022. We better explained it in the text.
Line 307. This part is not related to the rest of the paper. Why is it included in? More information in the previous parts of the paper is needed to include this conclusion.
R. During biological characterization of ToLCNDV-Le we observed a recovery phenotype in some plants of the host range (reference 31). In lines 306-317 of the Manuscript we discussed about the possibility that such a recovery phenotype could be the outcome of the Rbp interesting the AC1-AC4 region; the latter being a virus-coded suppressor of RNA silencing. Therefore, we think this part is a logical complement of the general discussion concerning the results of this study.
Line 329. More information about the origin of the samples should be given. Just one sample? No other samples isolated from other crops or fields? More samples should be included in the study.
R. This study focuses on the analysis of genome of a ToLCNDV isolate found in a single plant of zucchini grown in the Province of Lecce (southern Italy). Actually, ToLCNDV infection was widespread in the greenhouse crop and after confirmation of ToLCNDV presence by dot-blot hybridization with a Digoxigenin-labeled (DIG) DNA (reference 31), a single plant was selected to obtain the ToLCNDV-Le isolate, which has been characterized biologically (reference 31) and its genome sequence determined (this study).
Samples were from Lagenaria? Why not from cultivated species of Cucurbita?
R. We thank the Reviewer for evidencing this discrepancy. Actually we prepared nucleic acid for sequencing from the zucchini plant with natural infection. However, plant tissues were in a vary bad condition and as a result the external sequencing service informed us that the quality of nucleic acid preparation was very poor and non suitable for sequencing. Thus, ToLCNDV-Le was transferred to and maintained in a number of herbaceous hosts grown in laboratory greenhouses. Lagenaria siceraria was a very susceptible host; thus ToLCNDV-Le genome sequence was obtained after a single rub-inoculation on this plant that was used as source for nucleic acid preparation. We evidenced this in the 2.1 paragraph “Sequence analysis”.
Line 354. How many plants were used from Zucchini? All from the same field? Different isolates? The three samples from Lagenaria are from different fields? Different isolates?
R. Please see previous response. The three plants of Lagenaria siceraria did not came from naturally infected field but were rub-inoculated with sap extracted by the naturally infected zucchini plants from which the ToLCNDV-Le isolate was obtained (reference 31).
We hope that all the explanations reported above are good enough to proceed with the positive evaluation of our manuscript for publication in Plants.
With many thanks for your attention and time.
Please address all correspondence concerning this manuscript to me.
Best regards,
Dr. Tiziana Mascia (PhD)
Round 2
Reviewer 2 Report
The main aim of the paper is the molecular characterization of a tomato leaf curl New Delhi virus (ToLCNDV) isolate, denoted ToLCNDV--‐‑Le.
The paper shows an interesting information about the isolate, but the number of samples is unknown and if just one sample has been analysed the study has a serious limitation. More samples should be included to give a clue about the extent of this isolate. And more information is needed in some parts of the paper. This should be cleary explained in the paper and explain the limited scope of the study
Specific comments
Line 99. This paragraph gives results and conclusions. Here should the objective of the paper, not the results. Even it has been rewritten still results are given.
Line 151. In material and methods 120 samples are reported. In line 398 120 isolates is said to be used. Which is the total number?
Line 307. This part is not related to the rest of the paper. Why is it included in? A reference is given from a previous study but is the same isolated? As one sample was used in this study no relation can be given.
Line 336. More information about the origin of the samples should be given. Just one sample? No other samples isolated from other crops or fields? More samples should be included in the study. Samples were from Lagenaria? Why not from cultivated species of Cucurbita? Explanations are given in the authors reply but they do not appear in the paper.
Line 362. How many plants were used from Zucchini? All from the same field? Different isolates? The three samples from Lagenaria are from different fields? Different isolates? Explanations are given in the authors reply but they do not appear in the paper.
Author Response
Dear Reviewer,
thank you for your suggestions to improve the Manuscript. Please find below in red our answers to your comments.
The main aim of the paper is the molecular characterization of a tomato leaf curl New Delhi virus (ToLCNDV) isolate, denoted ToLCNDV-Le.
The paper shows an interesting information about the isolate, but the number of samples is unknown and if just one sample has been analysed the study has a serious limitation. More samples should be included to give a clue about the extent of this isolate. And more information is needed in some parts of the paper. This should be cleary explained in the paper and explain the limited scope of the study.
R. At the end of the Introduction (lines 91-97) we explained that the aim of this study was to determine the primary structure of DNA-A and DNA-B of the ToLCNDV-Le isolate. In other parts of the text we specified that this isolate was obtained from a single plant of zucchini squash (lines 110-115, 337-345 and 439-441). We hope these specifications solve any misunderstanding about the aim of the study and the number of plants analyzed.
Specific comments
Line 99. This paragraph gives results and conclusions. Here should the objective of the paper, not the results. Even it has been rewritten still results are given.
R. We rewrote the paragraph deleting an anticipation of the results, as suggested.
Line 151. In material and methods 120 samples are reported. In line 398 120 isolates is said to be used. Which is the total number?
R. We rewrote sections 2.2 and 4.4 to better explain the number of isolates used in this work.
Line 307. This part is not related to the rest of the paper. Why is it included in? A reference is given from a previous study but is the same isolated? As one sample was used in this study no relation can be given.
R. We specified (lines 323-326) the ToLCNDV-Le isolate was characterized biologically in the previous study and molecularly in this study. Thus, the isolate was the same and we think the relation is functional to discuss the results with the AC4 protein structure.
Line 336. More information about the origin of the samples should be given. Just one sample? No other samples isolated from other crops or fields? More samples should be included in the study. Samples were from Lagenaria? Why not from cultivated species of Cucurbita? Explanations are given in the authors reply but they do not appear in the paper.
R. We explained this in several parts of the manuscript as requested (lines 110-115, 129-132, 337-345, 372-375 and 439-441).
Line 362. How many plants were used from Zucchini? All from the same field? Different isolates? The three samples from Lagenaria are from different fields? Different isolates? Explanations are given in the authors reply but they do not appear in the paper.
R. We used one plant of naturally infected zucchini to transfer the inoculum to a number of herbaceous hosts. One plant of Langenaria siceraria was used to prepare nucleic acid for WGS whereas three rub-inoculated plants of L. siceraria were used to search for betasatellites. We explained this in several parts of the manuscript as requested (lines 110-115, 129-132, 337-345, 372-375 and 439-441).
We hope that the further suggestions fixed in the text to improve the Manuscript (ID plants-2427632) are good enough to proceed with the positive evaluation of our manuscript for publication in Plants.
With many thanks for your attention and time.
Best regards,
Dr. Tiziana Mascia
Round 3
Reviewer 2 Report
The paper shows an interesting information about the isolate, but the number of samples is unknown and if just one sample has been analysed the study has a serious limitation. The authors explained in the paper and explained the limited scope of the study
Specific comments
Line 317. This part is not related to the rest of the paper. Explanations given by the author do not explain why is important in this paper this information as is not related to the results of the paper. It should be removed.
Author Response
Dear Reviewer,
thank you for your message.
Please find below in red our answers to your comment for the Manuscript (ID plants-2427632) entitled “Molecular characterization of a recombinant isolate of tomato leaf curl New Delhi virus associated with severe outbreaks in zucchini squash in southern Italy” by Mariarosaria Mastrochirico, Roberta Spanò, Rita Milvia De Miccolis Angelini, Tiziana Mascia.
Reviewer 2
Comments and Suggestions for Authors
The paper shows an interesting information about the isolate, but the number of samples is unknown and if just one sample has been analysed the study has a serious limitation. The authors explained in the paper and explained the limited scope of the study.
Specific comments
Line 317. This part is not related to the rest of the paper. Explanations given by the author do not explain why is important in this paper this information as is not related to the results of the paper. It should be removed.
R.Removed as required
We hope that this deletion makes the Manuscript (ID plants-2427632) good enough to proceed with the positive evaluation of our manuscript for publication in Plants.
With many thanks for your attention and time.
Best regards,
Dr. Tiziana Mascia (PhD)
Round 4
Reviewer 2 Report
Changes have been made